# Healthcare Utilization for Lateral Epicondylitis: A 9-Year Analysis of the 2010–2018 Health Insurance Review and Assessment Service National Patient Sample Data

**DOI:** 10.3390/healthcare10040636

**Published:** 2022-03-28

**Authors:** Yujin Cho, Jiyoon Yeo, Ye-Seul Lee, Eun Jung Kim, Dongwoo Nam, Yeon-Cheol Park, In-Hyuk Ha, Yoon Jae Lee

**Affiliations:** 1Jaseng Hospital of Korean Medicine, 536 Gangnam-daero, Gangnam-gu, Seoul 06110, Korea; yujin6036@jaseng.co.kr; 2Jaseng Spine and Joint Research Institute, Jaseng Medical Foundation, 3F, 538 Gangnam-daero, Gangnam-gu, Seoul 06110, Korea; jyyeo0605@gmail.com (J.Y.); yeseul.j.lee@gmail.com (Y.-S.L.); hanihata@gmail.com (I.-H.H.); 3Department of Acupuncture & Moxibustion, College of Korean Medicine, Dongguk University, Gyeongju 13601, Korea; hanijjung@naver.com; 4Department of Acupuncture & Moxibustion, College of Korean Medicine, Kyung Hee University, 26 Kyungheedae-ro, Dongdaemun-gu, Seoul 02447, Korea; hanisanam@daum.net; 5Department of Acupuncture & Moxibustion, College of Korean Medicine, Kyung Hee University Korean Medicine Hospital at Gangdong, 892 Dongnam-ro, Gangdong-gu, Seoul 05278, Korea; icarus08@hanmail.net

**Keywords:** lateral epicondylitis, HIRA claims data, national patients sample, cost of care, medical service utilization

## Abstract

This retrospective cross-sectional study examined healthcare utilization among 213,025 patients with lateral epicondylitis over a nine-year period using the 2010–2018 Health Insurance Review and Assessment Service (HIRA) data (ICD code M771). Healthcare utilization, types of treatment, and the route of the visit were analyzed with frequency analysis for Western medicine (WM) and Korean medicine (KM). The findings revealed that the number of patients visiting WM and KM facilities for lateral epicondylitis rose every year from 2010 to 2018. Over this period, the age distribution of patients was 45–54 years (39.93%), 55–64 (23.12%), and 35–44 years (21.07%), and there were slightly more female patients (53.66%) than male patients (46.34%). The number of claims for lateral epicondylitis tended to increase with decreasing average monthly temperature; an increased proportion of middle-aged patients (45–64 years) was the most evident. The most frequently performed interventions in WM were subcutaneous or intramuscular injection (injection), deep heat therapy (physical therapy), and spinal peripheral nerve block-axillary nerve block (treatment/operation); the most frequently performed intervention in KM was acupuncture (injection). For pharmacological treatment, analgesics and anti-inflammatory medications were most frequently prescribed. The findings can be useful for health policymakers and as foundational data for clinicians and researchers.

## 1. Introduction

Lateral epicondylitis (ICD code M771) or tennis elbow is a condition in which active and resistance motion of extensors of the forearm exacerbates pain in the lateral epicondyle area [1]. Lateral epicondylitis is caused by a repetitive strain of extensor tendons, especially the tendon attached to the extensor carpi radialis brevis, or strong extension of or direct trauma to the lateral epicondyle [2]. Originally, the pathological process was explained as inflammation, but currently, the consensus is that microtrauma triggers a degenerative process [3]. Most patients have pain in the anterior or medial aspect of the upper part of the lateral epicondyle, and the pain usually radiates along the common extensors. The manifestation of pain ranges widely, from intermittent and mild pain to continuous and severe pain to the point of causing sleep disturbance [4].

Lateral epicondylitis can be caused by playing tennis or similar racquet sports, and it may occasionally be triggered by other types of sports or occupational activities [5]. Pain from lateral epicondylitis affects daily motions, such as moving, lifting, and grasping objects [6]. Some studies proposed that factors involving force, posture, and repetition as well as vigorous physical activities may also be associated with lateral epicondylitis [7,8]. Mental and psychological factors, such as depressive symptoms, have also been linked to lateral epicondylitis [8,9]. In addition, the condition may be aggravated by temperature, and smoking and obesity are some of the known risk factors [4,10,11].

The prevalence of lateral epicondylitis has been reported to range from 1 to 3% in the general population [12], and approximately one million cases are newly diagnosed every year in the United States [13]. In terms of age, people aged 40 years and over are more commonly affected [12]. In particular, the prevalence has been reported to be higher among people aged 40–49 years, followed by 50–59 years [13]. In a Finnish study on people aged 30–64 years, the prevalence was also higher among those aged 45–54 years [10]. In terms of sex, there were no marked differences between sexes [12], with the prevalence being reported to be 1–1.3% in men and 1.1–4.0% in women [14].

Lateral epicondylitis is diagnosed mostly based on the patient’s clinical history and physical examination [15]. Imaging is usually only performed when it is necessary to assess the severity of tissue injury and eliminate other causes [14]. Symptoms are resolved with rest and time in most cases, so conservative treatment is the treatment of choice in the early stages [5]. Initial conservative treatments include behavioral modification, non-steroidal anti-inflammatory drugs (NSAIDs), a strap, braces, physical therapy, extracorporeal shock wave therapy, injection, and laser therapy; and nonsurgical treatments are about 90% effective [1,4,16,17]. Patients who do not respond to six to nine months of conservative treatment are recommended to undergo imaging and may ultimately require surgery [5].

Due to the specific healthcare system in Korea, which provides dual healthcare divided into Korean Medicine and Western Medicine, acupuncture is provided by Korean Medical Doctors (KMD) who are licensed as Medical Doctors but specialize in diagnoses supplemented by medical knowledge based on Korean Medicine and therapies such as acupuncture, moxibustion, Tuina, and herbal medicine [18]. Although they are under the “dual” healthcare system, both are reimbursed by the National Health Insurance Service (NHIS) of Korea. Korean medicine (KM) conservative treatments such as acupuncture and moxibustion have been reported to be safe and effective for short-term pain relief for lateral epicondylitis [6,19,20,21,22,23,24,25,26].

The disease burden of lateral epicondylitis is on the rise every year. In a 2018 US study, the total insurance reimbursement for lateral epicondylitis treatments between 2007–2014 was $7,220,912, with per-patient reimbursement of $4263, and both costs were reported to increase annually [27]. Lateral epicondylitis induces functional impairment and incurs substantial costs related to productivity loss and healthcare utilization, and a considerable percentage of workers in an early stage of functional impairment have been reported to show a loss of productivity [14,17]. According to the 2010–2018 Health Insurance Review and Assessment Service (HIRA) disease-specific statistics in Korea, the number of patients visiting a Western medicine (WM) healthcare facility for lateral epicondylitis increased by 191,904 (approximately 41%) over nine years, while the total cost of care at WM healthcare facilities increased by $31,741,584.7 (approximately 104%) over the same period. Conversely, the number of patients visiting a KM healthcare facility increased by 74,820 (approximately 105%) over nine years, with the total cost of care increasing by $10,104,087.17 (approximately 225%) in the same period [28].

Given the above, there is a need to examine the latest healthcare utilization trends for lateral epicondylitis amid the growing patient population and cost of care for this condition, and additional studies are needed to optimize the treatment algorithm in order to reduce relevant medical costs [27]. A 2016 US population-based study on healthcare utilization and direct medical cost using a cohort of patients who received treatment for lateral epicondylitis at least twice reported that the greatest annual direct medical cost was incurred from examination (32%), followed by physical therapy (23%), and surgery (20%) [29]. In 2018, a US study analyzed the epidemiology and the cost of the disease burden of lateral epicondylitis using health insurance data [27].

In Korea, one study analyzed 148 cases of lateral epicondylitis to examine the epidemiology, assess conservative treatment, and assess surgical treatment [30]. Further, other studies investigated the effectiveness of extracorporeal shock wave therapy (ESWT), prolotherapy, platelet-rich plasma (PRP) injection, and arthroscopy [31,32,33,34], as well as case reports, systematic reviews, and randomized controlled trials (RCTs) on acupuncture (general acupuncture, pharmacopuncture, bee venom acupuncture, and burning acupuncture) [35,36,37,38,39,40]. However, studies shedding light on healthcare utilization and cost of care for lateral epicondylitis in Korea are lacking. Moreover, as South Korea features a dual healthcare system, additional studies are needed to compare and analyze treatments in both WM and KM comprehensively. Thus, national-level studies that analyze the types of treatments, cost, and frequency of treatment performed in WM and KM are required to examine healthcare utilization for lateral epicondylitis in the context of the dual healthcare system in Korea.

Against this background, this study comparatively analyzed the current state of lateral epicondylitis, patient characteristics, and health service categories and utilization for this condition in WM and KM using the HIRA claims data from 2010–2018. Ultimately, we aim to provide comprehensive and universal baseline data useful for developing health policies pertaining to reducing the healthcare burden and establishing treatment guidelines for lateral epicondylitis.

## 2. Materials and Methods

### 2.1. Data Source

This study used the 2010–2018 HIRA-National Patient Sample (NPS) data. Health insurance claims data are generated when healthcare providers submit their claims for fee reimbursement at HIRA for the services provided to patients. HIRA data are highly useful for healthcare research, as it contains information about patients’ diagnosis, treatment details, procedures, surgical history, and prescribed medications [41]. The patient sample data are provided as secondary data after removing information about individuals and legal entities from the raw data and randomly stratified sample data in one-year units. The data contains details of patient care and prescription claims for one year from the date of treatment initiation for the corresponding year and are stratified by sex and age groups (10-year units).

### 2.2. Study Population and Design

This study was conducted on patients with lateral epicondylitis (KCD-10 code M771) as the primary diagnosis during the study period, and patients of all ages who had received healthcare service at least once with KCD-10 Code M771 as the primary diagnosis. The KCD code is the adopted version of the ICD code in Korea with a few changes to reflect the clinical settings in Korea [18]. While this included chronic symptoms on the elbow due to overuse with repetitive motions on the elbow, such as the “Tennis elbow”, the definition did not include acute elbow injuries by trauma defined using S codes. Cases claimed by dental facilities, public health facilities, and psychiatric facilities (1134 cases), cases with an institution code for long-term care hospitals, psychiatric long-term care hospitals, dental hospitals, maternity centers, and public health facilities (2835 cases), and cases with total cost and number of days in care recorded as 0 or missing (394 cases) were excluded, and a total of 759,347 claims for 213,025 patients were included in our analysis.

### 2.3. Statistical Data Analysis

In this study, healthcare utilization for lateral epicondylitis was examined separately for WM and KM and as overall, and the number of patients, total claims, total expense, per-patient expense, and per-claim expense were analyzed.

The selected patients were analyzed by age (10-year units from <15 years to ≥75 years; eight categories), sex, payer type (national health insurance, Medicaid, patriot and veteran benefit). Further, the claims were classified into the type of visit (inpatient, outpatient) and type of healthcare facility (tertiary hospital/general hospital/hospital, clinic, KM hospital, KM clinic), and the frequencies and percentages were analyzed.

To analyze the seasonal trends of lateral epicondylitis, the increase in the number of claims for this condition according to average monthly temperature and rate of increase by age group over the nine-year period were analyzed.

To analyze the healthcare services provided for patients with lateral epicondylitis, the service categories were divided into examination, medication administration, physical therapy, treatment/operation, test, and diagnostic radiology per the Ministry of Health and Welfare (MOHW) criteria, and the average annual expense and the average annual number of claims were calculated for the nine-year period. The average annual log change rate was calculated under the assumption that the frequency of each item rose at a steady rate over the years, and the change rates were presented for each service category in WM and KM.

To identify frequently performed interventions in WM and KM facilities, the service codes under each medical service category covered by NHIS were reviewed. Further, the five most frequently performed interventions for physical therapy, injection, and treatment/operation categories in WM and that for the injection category in KM were listed in a table.

For patients who initially visited a KM facility for lateral epicondylitis, switched to WM, and returned to a KM facility (KM-WM-KM group), the procedure codes in the claims submitted by the WM facility were reviewed to identify the reasons of their choices.

Drugs prescribed (outpatient pharmacy and in-hospital prescription) for lateral epicondylitis were classified according to the ATC-code 2nd level criteria with reference to the MOHW classification criteria, and the frequencies of prescription of each category during the nine-year period were presented in a table.

All expenses were converted to USD based on the average KRW/USD exchange ratio for the corresponding year and adjusted based on the consumer price index for the health sector in 2018. All statistical analyses were performed using the SAS 9.4 TS Level 1M4 (2002–2012 by SAS Institute Inc., Cary, NC, USA) software.

## 3. Results

### 3.1. Number of Patients and Cost of Medical Care for Lateral Epicondylitis

According to Table 1, a total of 16,673 patients visited a healthcare facility for lateral epicondylitis in 2010, with 14,319 utilizing WM and 2354 utilizing KM. In 2018, the total number of patients rose by about 45%, with a 39% increase in WM users and an 87% increase in KM users. Further, the number of WM users was approximately six times greater than that of KM users in 2010, but the gap narrowed year after year to about 3.6-fold in 2014; however, the gap again widened since 2015 to approximately 4.5-fold in 2018. The total number of claims in 2010 was 63,649, with 53,704 claims in WM and 9945 claims in KM. In 2018, the total number of claims was 96,301, with 76,256 claims in WM and 20,045 claims in KM. While the total number of claims and WM claims rose steadily, the number of KM claims rose until 2015 but began to decline in 2016. Total expenses rose by about 131% from 1,027,367 USD in 2010 to 2,377,540 USD in 2018. The WM expense and KM expense rose from 883,304 USD and 144,063 USD, respectively, in 2010 to 1,913,038 USD and 464,501 USD, respectively, in 2018. Per-patient expense in WM rose from 62 USD to 96 USD in 2018, and that in KM rose from 61 USD to 106 USD in 2018. Between WM and KM, the per-patient expense in WM was slightly higher than that in KM in 2010, but the per-patient expense in KM remained higher since 2011. However, the per-claim expense was higher in WM than KM every year during the nine-year period. The greater number of visits among KM users would have led to the greater annual per-patient expense in KM despite higher annual per-claim expense in WM.

### 3.2. Characteristics of Patients with Lateral Epicondylitis

According to Table 2, more female patients visited a healthcare facility for lateral epicondylitis (96,258; 53.66%) than male patients (83,136, 46.34%). Among WM users, the percentages of male and female patients were 46.09% and 53.91%, respectively, and among KM users, the percentages were 47.78% and 52.22%, respectively. The predominant age group of patients utilizing healthcare for lateral epicondylitis was 45–54 years (39.93%), followed by 55–64 years (23.12%) and 35–44 years (21.07%), and the order of most common age group was consistent among both WM users and KM users. The insurance type was NHI (97.24%), followed by Medicaid (2.72%), and there were no marked differences in the insurance types between WM and KM.

### 3.3. Medical Usage of Patients with Lateral Epicondylitis

Appendix A shows that most cases of treatment for lateral epicondylitis occurred through outpatient services (99.78%) as opposed to inpatient services (0.22%). In WM, 81.37% of the treatments were given at a primary care facility (clinic), with 18.58% of the treatments given at a tertiary hospital/general hospital/hospital, showing that about 4/5 of the care for lateral epicondylitis is given at a primary care facility. In KM, 98.08% of the treatments were performed at a primary care facility (KM clinic), with 1.82% of the treatments performed in a KM hospital, showing a markedly high utilization of primary care facilities.

### 3.4. Number of Patients with Lateral Epicondylitis According to Average Monthly Temperature and Trends by Age during a Nine-Year Period

Figure 1 shows a graph of the average monthly temperatures and the number of newly diagnosed patients in Korea in the nine-year period. In general, the number of patients increased with decreasing average monthly temperature, and the number of patients decreased with increasing average monthly temperature. Particularly, the number of patients tended to peak in December, when the temperature drops significantly compared to the preceding month (Appendix A). Regarding age-related trends, the proportion of 45–54-year-olds and 55–64-year-olds tended to increase steadily every year from 2010–2018, while the proportion of ≤44-year-olds and ≥65-year-olds did not increase noticeably.

### 3.5. Average Rate of Change of Total Expense and Number of Claims by Category

As shown in Table 3, the service category that incurred the highest annual average expense in the nine-year period was examination (769,713.08 USD), with an average 8.52% increase annually and an average of 115,246.67 cases per year. The next highest expense was for injection (284,659.14 USD), with an average 13.48% increase annually and an average of 113,102.78 cases per year. The third-highest expense was for physical therapy (211,598.24 USD), with an average 2.86% increase annually and an average of 117,633.56 cases per year. In WM, the service category with the highest cost was examination (637,306.34 USD), followed by physical therapy (211,598.24 USD), treatment/operation (209,533.05 USD), diagnostic radiology (111,718.35 USD), and injection (68,973.14 USD). The service category with the highest number of claims was physical therapy (117,633.56 cases), followed by examination (95,808.89 cases), injection (44,653.67 cases), diagnostic radiology (19,608.56), and treatment/operation (14,882.56). In WM, the service category with the highest average change rate (19.77%) over nine years was treatment/operation. In KM, the service category with the highest cost was injection (215,686.00 USD), followed by examination (132,406.75 USD), and medication administration (3000.48 USD), and that with the highest number of claims was also injection (68,449.11 cases) followed by examination (19,437.78 cases), and medication administration (3521.78 cases), with an average change rate of 15.45%. Physical therapy, diagnostic radiology, and treatment/operation were not collated because they are not separately billed in KM facilities.

### 3.6. Top Five Frequently Performed Interventions in WM and km over Nine Years

Table 4 shows the five most frequently performed interventions in each service category in WM and KM. The physical therapy, injection, and treatment/operation categories in WM and the injection category in KM (because all interventions are billed as an injection in KM) were reviewed. The most frequently performed intervention in the WM-physical therapy category was deep heat therapy (DHT), with 308,478 claims, per-claim expense of 1.03 USD, and per-patient expense of 3.98 USD, where patients are estimated to have received the intervention 3.86 times a year on average. In the WM-physical therapy category, the intervention with the highest per-claim expense was low-level laser therapy (LLLT; 5.02 USD), and that with the highest per-patient expense was also LLLT (16.23 USD). In the WM-injection category, the most frequently performed intervention was subcutaneous or intramuscular injection (SC or IM Inj), with 114,227 claims, per-claim expense of 1.12 USD, and per-patient expense of 2.45 USD, where patients are estimated to have received the intervention 2.18 times a year on average. The number of claims for SC or IM Inj was 4.56 times higher than that for intratendinous injection (IT Inj).

In the WM- injection category, the intervention with the highest per-claim expense was intraarticular injection (IA Inj; 13.88 USD), and that with the highest per-patient expense was also IA Inj (20.46 USD). In the WM-treatment/operation category, the most frequently performed intervention was spinal peripheral nerve block-medial nerve block, ulnar nerve block, and radial nerve block (MNB, UNB, RNB), with 49,297 claims, per-claim expense of 18.95 USD, and per-patient expense of 46.45 USD, where patients are estimated to have received the intervention 2.45 times a year on average. In the WM-treatment/operation category, the intervention with the highest per-claim expense was splint (21.46 USD), and that with the highest per-patient expense was MNB, UNB, RNB (46.45 USD).

In the KM-injection category, the most frequently performed intervention was acupuncture (Acu), with 313,109 claims, per-claim expense of 3.58 USD, and per-patient expense of 29.84 USD, where patients are estimated to have received the intervention 8.33 times a year on average. The second most frequently performed intervention was infrared therapy (IR; 71,772 claims), followed by cupping (67,928 claims), electroacupuncture (EA; 48,241 claims), and indirect moxibustion (Indirect Moxa; 35,264 claims). The number of claims for Acu was about 4.36 times higher than that for IR. In the KM-injection category, the intervention with the highest per-claim expense was cupping (4.46 USD), and that with the highest per-patient expense was Acu (29.84 USD).

### 3.7. Healthcare Utilization in WM among Patients Who Switched between KM and WM

Table 5 shows the details of WM treatments provided for patients who were first diagnosed with lateral epicondylitis at a KM facility, subsequently sought WM care, and switched back to KM care (KWK+ patients). The most frequently billed service category was physical therapy (45.4%), followed by examination (35.05%), injection (5.93%), and diagnostic radiology (5.91%). Among patients who returned to the initial KM facility within seven days after diagnosis, 14.33% received a radiologic diagnosis at a WM facility before returning to the KM facility. Among those who returned to the initial KM facility within 14 days of initial diagnosis, 10.80% received a radiologic diagnosis at a WM facility before returning to the KM facility. The rate of radiologic diagnosis was 5.91% in the KWK+ group, but the rate was about 2.42-fold higher (14.33%) in the group that returned to the same KM facility within seven days of an initial visit.

## 4. Discussion

This study analyzed healthcare utilization for lateral epicondylitis, including the number of patients with this condition, their medical expenditure and characteristics, route of visit to the healthcare facility, and frequently performed interventions using the 2010–2018 HIRA-NPS data. The total number of patients with lateral epicondylitis, the total number of claims, and total medical expenditure are growing year after year. Compared to that in 2010, the number of patients visiting a healthcare facility for lateral epicondylitis increased by approximately 45.43%, and total expenditure increased by about 131.42% in 2018. These results are in line with previous findings that the medical expenditure for lateral epicondylitis had been on an increasing trend between 2007–2014 [27]. Moreover, these findings highlight the need for specific guidelines for lateral epicondylitis and other than the guidelines for elbow diseases published by the American College of Occupational and Environmental Medicine, as studies attempting to develop guidelines exclusively for lateral epicondylitis are generally lacking [42].

Regarding the sex and age-related trends of patients seeking healthcare for lateral epicondylitis in the specified nine-year period, the number of female patients (96,258, 53.66%) was about 1.15 times greater than that of male patients (83,136, 46.34%). This is consistent with a previous report that the prevalence of lateral epicondylitis is slightly higher among women (1.1–4.0%) than men (1–1.3%) [14]. The most commonly affected age group was 45–54 years (39.93%), followed by 55–64 years (23.12%) and 35–44 years (21.07%), consistent with the report that lateral epicondylitis most commonly affects people aged 40 years and over [12]. In particular, our results are in line with the report that the prevalence of lateral epicondylitis is higher in ages 40–49 years [13] and with the results of a Finnish study on people aged 30–64 years that the prevalence is higher in 45–54-year-olds [10].

In terms of the types of healthcare facilities, most treatments were performed at a primary care facility. In WM, the rate of utilization was 18.58% for tertiary hospital/general hospital/hospital and 81.37% for clinics, showing that 4/5 of WM treatments are performed at a primary care facility. In KM, the rate of utilization was 1.82% for KM hospitals and 98.08% for KM clinics, showing that KM treatments are predominantly performed at a primary care facility. Moreover, most of the treatments were performed through outpatient services (99.78%), as opposed to inpatient services (0.22%), suggesting that lateral epicondylitis is mostly treated with conservative treatments that are performed in outpatient settings than treatments that require hospitalization, such as surgery. This is consistent with previous findings that most cases of lateral epicondylitis are managed and primarily treated through conservative approaches [43,44].

Regarding the association between temperature and lateral epicondylitis, the number of patients tended to increase with decreasing average monthly temperature, and vice versa (Figure 1). In particular, the number of patients peaked in December—a winter month with a low average temperature in Korea, a country in the northern hemisphere. The reason for this may be that cold weather tends to exacerbate joint pain [11]. One report showed that the incidence of injuries in cold weather has been rising in the past 20 years as a result of increased outdoor leisure and sports activities [45]. Cold weather slows all chemical processes in the body, including nervous system activities that trigger muscular contraction, thereby elevating the risk for injuries [46]. Hence, individuals should be particularly careful with using their elbows during cold weather.

From 2010–2018, the proportion of patients aged 45–54 years and 55–64 years has risen steadily every year, while the percentage of those under 44 years remained relatively unchanged. Previous studies reported that lateral epicondylitis most commonly affects people aged 40–59 years [23,47,48] and that its incidence is 2–3.5 times higher in the ≥40 years group than the <40 years group [49]. It has also been reported that the greatest cause of lateral epicondylitis among middle-aged adults (40–60 years) is weakened tendons or diminished tendon elasticity [49]. Although an injury occurs through a similar mechanism in late middle-aged adults (50–64 years) from that in young adulthood, the outcome or site of injury is similar to that in older adulthood, with increased frequency of injuries during leisure activities and trips [50]. Late middle-aged adults are at an elevated risk for injury and are vulnerable to more severe injuries due to their desires to maintain the same level of activities despite their physical changes, diminished vigor, and increased comorbidities [51]. Thus, middle-aged adults should be the priority target in promoting efforts to ensure early prevention and precautions.

As shown in Table 3, the service category in WM with the highest annual average expenditure over nine years was physical therapy, followed by treatment/operation, diagnostic radiology, and injection. Physical therapy was ranked first in both expense and number of claims. The annual average expense over nine years was about threefold higher for treatment/operation than an injection, while the annual average number of claims over nine years was about threefold higher for injection than treatment/operation. These results are similar to those showing that the annual expense pertaining to the treatment of lateral epicondylitis is the highest for physical therapy (23%), followed by surgery (20%), injection (4%), and diagnostic radiology (4%) [29]. However, our results differed in that the average expense was 1.6 times higher for diagnostic radiology than injection. In KM, both the average expense and number of claims over nine years were the highest for the injection category, followed by examination and medication administration. In KM, acupuncture, cupping, and moxibustion therapies are all billed as an injection, which seems to be the reason for the high utilization of injection.

As shown in Table 4, the most commonly performed intervention in the WM-physical therapy category was DHT, followed by SHT, TENS, ICT, and LLLT. Ultrasound produced a substantial short-term pain relief for lateral epicondylitis [52,53,54], and combined microwave therapy and exercise therapy was effective in pain relief and functional improvement for this condition [55]. TENS is generally used to alleviate pain associated with lateral epicondylitis [56,57]. ICT is a useful conservative treatment for lateral epicondylitis [58] and has been reported to be effective in relieving pain, resolving functional impairment, and increasing grip strength but less effective than ESWT [59]. While laser therapy has been reported to be more effective than braces in producing pain relief for lateral epicondylitis [60], other studies reported that no marked differences in pain and grip strength were observed between the laser therapy group and placebo group [54,61].

In the WM-injection category, SC or IM Inj was the most commonly performed intervention. Intramuscular injections were reported to be effective and safe for lateral epicondylitis, with less pain around the injection site [62]. Additionally, studies have suggested that steroid injection has a short-term effect in relieving pain and increasing grip strength but that drug therapy or physical therapy are more beneficial in the long term [63,64]. One study also reported that rehabilitation programs, and not steroid injections, should be first-line therapy [65]. Autologous blood injection (ABI) was effective in the short term but is not beneficial in the long term, while autologous PRP injection improved pain and functions but had no clear benefits over steroid injection, ABI, and saline injection [17]. One report showed that regenerative injections, such as ABI, PRP, and prolotherapy, were more effective in producing long-term pain relief than steroid injections [66].

In the WM-treatment/operation category, the most frequently performed intervention was MNB, UNB, RNB. It was previously reported that an ultrasound-guided posterior antebrachial cutaneous nerve (PABCN) block produced pain relief in patients with chronic lateral elbow pain and thus that it has diagnostic and therapeutic implications [67]. Surgical treatment can be an alternative for patients with chronic pain or functional impairment not controlled, even with appropriate nonsurgical management [17].

In the KM-injection category, the most frequently performed intervention was Acu, followed by IR, cupping, EA, and Indirect Moxa. The US National Institutes of Health suggested that Acu may be a good alternative for the treatment of lateral epicondylitis [68], and it has been found to be effective in producing short-term pain relief [6,17]. Acu was also most effective than the placebo for reducing pain and improving arm functions in patients with chronic lateral epicondylitis [21]. IR can reduce inflammation and improve the rate of damaged tissue recovery [69]. The combination of cupping and EA was more effective in treating lateral epicondylitis [70], and combined Chuna and EA also produced better therapeutic effects [71]. In a study that compared Acu and EA, EA was found to be more effective in treating lateral epicondylitis [72]. The combination of Moxa and Acu was more effective than Acu alone in treating lateral epicondylitis [25]. From a number of guidelines mentioned above, it seems reasonable to recommend Acu to treat lateral epicondylitis as guideline-based therapy; however, further studies are necessary to validate the cost-effectiveness of Acu in lateral epicondylitis.

Table 5 shows that the rate of radiological diagnosis was about 2.42-fold higher in the group that returned to the same KM clinic within seven days of the initial visit than in the KWK+ group. While radiologic assessment is not required for the diagnosis of lateral epicondylitis in most cases, radiology can be used to assess the degree of tissue damage and differentiation from other diseases [14]. While one benefit of the dual healthcare system in Korea featuring two distinct healthcare systems (WM/KM) is that patients can choose their preferred healthcare system [73], one downside is that patients who preferred to be examined at a KM facility still may have to visit a WM facility for radiologic diagnosis for reasons such as differential diagnosis, which is not only inconvenient but also may incur additional medical costs.

Patients with lateral epicondylitis are generally prescribed nonsteroidal anti-inflammatory drugs (NSAIDs) or acetaminophen (AAP). Although topical NSAIDs produce short-term pain relief for lateral epicondylitis, the use of oral NSAIDs remains controversial [68]. The most frequently prescribed oral medications for lateral epicondylitis in WM were analgesics and anti-inflammatory agents, digestive medications, and muscle relaxants. The most frequently prescribed analgesic and anti-inflammatory agents were NSAIDs, followed by anti-inflammatory enzymes, and AAPs. The per-claim expense and per-patient expense were also the highest for NSAIDs (Appendix A).

### Limitations and Strengths

First, the study data contained nationwide healthcare utilization data pertaining to lateral epicondylitis in the corresponding year, but it is difficult to conclude that the patients only received treatment for it solely based on the diagnosis classification used by the HIRA. To compensate for this, we excluded patients for whom lateral epicondylitis was coded as a sub-diagnosis and only included patients with lateral epicondylitis as the primary diagnosis, but we could not completely isolate lateral epicondylitis from other diseases.

Second, our study data are not continuous in nature. In other words, the data are accumulated in one-year units, where a patient can be traced only within the same year and cannot be followed up to the subsequent year.

Third, in general, the cost of care is divided into direct cost, indirect cost, and intangible cost. However, we excluded indirect cost and intangible cost and excluded non-medical cost from direct cost in this study to analyze only direct insurance-covered medical cost. The number of reimbursement items for KM included in the claims data is extremely small compared to that of WM procedures, and a substantial portion of KM procedures is non-reimbursement items [28]. Hence, we cannot conclude that we have reflected the full details of KM treatment given for patients with lateral epicondylitis solely based on the KM claims data.

Despite these limitations, this study has several strengths. First, this study is the latest study analyzing healthcare utilization for lateral epicondylitis using nine years’ worth of HIRA-NPS data from 2010–2018. No prior study has compared healthcare utilization for this condition by year in a single country.

Second, this study is the first study examining the association between average monthly temperature and incidence of lateral epicondylitis as well as age trends for this condition by year.

Third, this study compared healthcare utilization in WM and KM to reflect the special context of Korea that features a dual healthcare system. This is also the first study to compare healthcare utilization between the two systems by analyzing frequently performed interventions for each service category, cost, and the number of claims.

Finally, this study analyzed the route of a visit to a healthcare facility by patients with lateral epicondylitis by 10-year age groups and by sex, which sheds light on the specific information about which gender and age groups utilize which type of healthcare facility.

## 5. Conclusions

This study reviewed healthcare utilization for lateral epicondylitis in Korea using nine years of HIRA-NPS (January 2010–December 2018). The number of patients with lateral epicondylitis and medical cost is on the rise overall, and physical therapy and acupuncture were the most frequently performed interventions in WM and KM, respectively. Based on previous findings on the effectiveness of acupuncture in treating lateral epicondylitis [67] and the level of acupuncture utilization, further consideration is warranted on including acupuncture as a standard recommendation in clinical guidelines, and further studies are necessary to confirm such decisions. Considering that national-level studies to compare healthcare utilization for lateral epicondylitis between WM and KM facilities in Korea were lacking, the findings of this study—showing details of patient characteristics, treatment types, and cost pertaining to lateral epicondylitis in WM and KM separately—can serve as a valuable reference for clinicians and researchers in establishing standard treatment protocol. Further, the findings will be useful as baseline data for health policymakers and experts in their national health policy decisions, such as determining health insurance fees and budget allocation.

## Figures and Tables

**Figure 1 healthcare-10-00636-f001:**
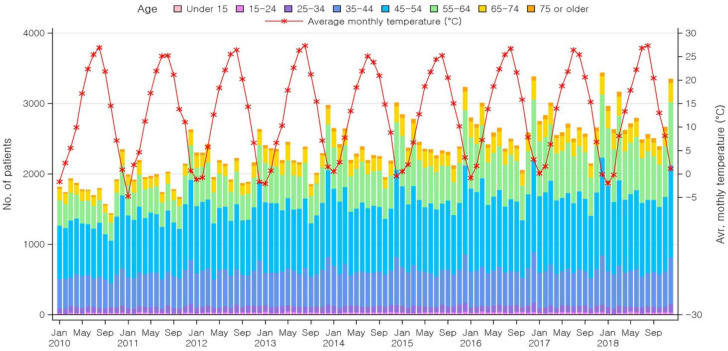
Number of lateral epicondylitis claims according to average monthly temperature and yearly changes in the number of patients by age group over nine years.

**Table 1 healthcare-10-00636-t001:** General medical service use for lateral epicondylitis patients in Korea.

Year	Type of Visit	Number of Patients	Total Claims	Total Expense	Per-Patient Expense	Per-Claim Expense
2010	Total	16,673	63,649	1,027,366.84	61.62	16.14
	WM	14,319	53,704	883,303.88	61.69	16.45
	KM	2354	9945	144,062.96	61.20	14.49
2011	Total	18,642	70,810	1,198,216.61	64.28	16.92
	WM	14,948	55,203	942,777.63	63.07	17.08
	KM	3694	15,607	255,438.98	69.15	16.37
2012	Total	20,008	78,832	1,322,261.65	66.09	16.77
	WM	15,908	61,229	1,028,142.16	64.63	16.79
	KM	4100	17,603	294,119.49	71.74	16.71
2013	Total	21,249	85,219	1,581,431.76	74.42	18.56
	WM	16,677	64,820	1,581,431.76	72.84	18.74
	KM	4572	20,399	366,649.45	80.19	17.97
2014	Total	21,676	86,659	1,763,041.28	81.34	20.34
	WM	16,932	66,025	1,352,926.69	79.90	20.49
	KM	4744	20,634	410,114.59	86.45	19.88
2015	Total	22,507	88,730	1,783,651.54	79.25	20.10
	WM	17,768	68,060	1,387,969.10	78.12	20.39
	KM	4739	20,670	395,682.44	83.49	19.14
2016	Total	23,657	93,377	1,897,337.27	80.20	20.32
	WM	18,993	72,813	1,494,260.04	78.67	20.52
	KM	4664	20,564	403,077.23	86.42	19.60
2017	Total	23,999	95,770	2,130,992.79	88.80	22.25
	WM	19,447	75,444	1,700,737.50	87.46	22.54
	KM	4552	20,326	430,255.29	94.52	21.17
2018	Total	24,248	96,301	2,377,539.57	98.05	24.69
	WM	19,851	76,256	1,913,038.13	96.37	25.09
	KM	4397	20,045	464,501.44	105.64	23.17

WM, Western Medicine; KM, Korean Medicine. All expenditures were converted based on the annual average exchange rate (KRW/USD) and the price was adjusted considering the health expenditure price level of the year 2018. (See Appendix A in detail).

**Table 2 healthcare-10-00636-t002:** Basic characteristics of patients.

Category		Patients							
		Total (2010–2018)		Western Medicine Only (2010–2018)		Korean Medicine Only (2010–2018)		Used Both (2010–2018)	
		No. of Patients	%	No. of Patients	%	No. of Patients	%	No. of Patients	%
Age	Younger than 15	605	0.34	516	0.36	84	0.34	5	0.04
	15–24	2453	1.37	2113	1.49	301	1.23	39	0.29
	25–34	7442	4.15	6095	4.31	1013	4.13	334	2.52
	35–44	37,806	21.07	29,673	20.96	4947	20.15	3186	24.02
	45–54	71,624	39.93	55,988	39.55	9598	39.09	6038	45.52
	55–64	41,483	23.12	33,248	23.48	5494	22.38	2741	20.66
	65–74	14,709	8.2	11,366	8.03	2522	10.27	821	6.19
	75-	3272	1.82	2579	1.82	592	2.41	101	0.76
Gender	Male	83,136	46.34	65,253	46.09	11,730	47.78	6153	46.39
	Female	96,258	53.66	76,325	53.91	12,821	52.22	7112	53.61
Payer type	NHI *	174,450	97.24	137,559	97.16	23,888	97.3	13,003	98.02
	Medicaid	4874	2.72	3949	2.79	663	2.7	262	1.98
	Others **	70	0.04	70	0.05	-	-	-	-

*** NHI, National Health Insurance. ** Nationally funded care/patriots and veterans.

**Table 3 healthcare-10-00636-t003:** Average expenditure † and total number of claims, average change rate over nine years.

	All				Western Medicine				Korean Medicine			
	Total Expenditure †		No. of Claims		Total Expenditure †		No. of Claims		Total Expenditure †		No. of Claims	
	Avr. Exp *	Avr. CR *	Avr. Exp *	Avr. CR *	Avr. Exp *	Avr. CR *	Avr. Exp *	Avr. CR *	Avr. Exp *	Avr. CR *	Avr. Exp *	Avr. CR *
Examination	769,713.08	8.52	115,246.67	5.43	637,306.34	7.79	95,808.89	4.75	132,406.75	13.09	19,437.78	9.73
Hospitalization	57,101.80	19.64	1165.33	17.72	55,569.18	19.39	1139.89	18.51	1532.62	57.01	25.44	−2.62
Medication administration (+ prescription)	10,763.33	13.99	7642.56	10.53	7762.84	11.42	4120.78	2.69	3000.48	22.82	3521.78	26.94
Physical therapy	211,598.24	2.86	117,633.56	0.78	211,598.24	2.86	117,633.56	0.78	-	-	-	-
Injection	284,659.14	13.48	113,102.78	6.50	68,973.14	9.17	44,653.67	1.44	215,686.00	15.45	68,449.11	10.88
Treatment/operation	209,533.05	19.77	14,882.56	11.17	209,533.05	19.77	14,882.56	11.17	-	-	-	-
Test	33,358.06	16.43	10,203.11	11.69	33,151.01	16.59	10,149.00	11.75	207.05	−5.32	54.11	−0.90
Diagnostic radiology	111,718.35	11.13	19,608.56	1.60	111,718.35	11.13	19,608.56	1.60	-	-	-	-

* Avg. exp, Annual average gross expenditure over nine years; Avg. N, Annual average number of cases over nine years; Avg. CR, Annual average log change rate over nine years. † All expenditures were converted with annual average exchange rate (KRW/USD). Price level of health expenditure was adjusted as of year 2018. (See Appendix A in detail).

**Table 4 healthcare-10-00636-t004:** The five most frequently performed interventions in WM/KM over nine years.

Category		Subcategory	Number of Claims	Average Cost per Claims	Average Cost per Patient
WM	Physical therapy	Deep Heat Therapy (DHT)	308,478	1.03	3.98
		Superficial Heat Therapy (SHT)	300,923	0.38	1.46
		Transcutaneous Electrical Nerve Stimulation (TENS)	185,116	3.09	10.87
		Interferential current therapy (ICT)	109,150	3.16	11
		Low Level Laser Therapy (LLLT)	54,598	5.02	16.23
	Injection	Subcutaneous or Intramuscular Inj.	114,227	1.12	2.45
		Intratendinous Inj.	25,041	4.45	6.74
		Perineural Inj.	6142	5.56	9.66
		Intraarticular Inj.	5831	13.88	20.46
		Intravenous Catheter	2541	0.98	1.46
	Treatment/operation	Median, Ulnar, Radial N-Block	49,297	18.95	46.45
		Axillary N-Block	5661	17.34	41.35
		Scapular N-Block	4548	19.25	39.63
		Dressing	4031	8.67	18.7
		Splint	2209	21.46	22.68
KM	Injection	Acupuncture	313,109	3.58	29.84
		Infra-red therapy	71,772	0.83	3.02
		cupping	67,928	4.46	14.99
		Electro Acupuncture	48,241	3.8	15.22
		Indirect Moxibustion	35,264	2.33	8.77

All expenditures were converted based on the annual average exchange rate (KRW/USD) and the price was adjusted considering the health expenditure price level of the year 2018. (See Appendix A in detail).

**Table 5 healthcare-10-00636-t005:** WM interventions received among patients who alternated between KM and WM institutions.

Category	WK+ Total		Returned within 7 Days		Returned within 14 Days	
	No. of Claims	%	No. of Claims	%	No. of Claims	%
Physical therapy	12,286	45.40%	395	40.14%	1133	43.24%
Diagnostic radiology	1599	5.91%	141	14.33%	283	10.80%
Examination	9486	35.05%	306	31.10%	859	32.79%
Injection	1604	5.93%	65	6.61%	180	6.87%
Testing	954	3.52%	60	6.10%	105	4.01%
Anesthesia	673	2.49%	10	1.02%	36	1.37%
Treatment/operation	36	0.13%	1	0.10%	2	0.08%
Others	426	1.57%	6	0.61%	22	0.84%

## Data Availability

The datasets generated during and/or analyzed during the current study are available in the HIRA-NPS repository upon request [http://opendata.hira.or.kr, accessed on 29 July 2021] and upon payment of a data request additional fee.

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
