# Peer review of "Healthcare Utilization for Lateral Epicondylitis: A 9-Year Analysis of the 2010–2018 Health Insurance Review and Assessment Service National Patient Sample Data"

_healthcare, 2022, doi:10.3390/healthcare10040636_

Round 1

Reviewer 1 Report

The authors conducted a retrospective cross-sectional study to investigate the utilization of medical services for lateral epicondylitis.
For this purpose, they use the health data of the so-called HIRA. In addition, they compare therapeutic approaches in Western and Korean medicine. The aim of the study is to establish baseline or epidemiological data regarding treatment and costs of lateral epicondylitis.

Introduction: 
They distinguish between Western and Korean medicine. Therapies such as acupuncture have already become established in Western medicine. For example, in Germany, acupuncture is used by specialists as a standard treatment for spinal diseases and gonarthrosis. The services are covered by health insurance companies. Is it therefore correct to divide therapy options into western and korean, or would a classification into different therapy approaches be possible?

Method:
You describe possible risk factors for epicondylitis. Are post-traumatic epicondylitis also considered?

Results:
Section 3.1. is overloaded with numbers for the reader. Please revise.

Discussion:

Based on your results, can you make recommendations for guideline-based therapy? Please justify.

Author Response

Reviewer 1:

Introduction:

They distinguish between Western and Korean medicine. Therapies such as acupuncture have already become established in Western medicine. For example, in Germany, acupuncture is used by specialists as a standard treatment for spinal diseases and gonarthrosis. The services are covered by health insurance companies. Is it therefore correct to divide therapy options into western and korean, or would a classification into different therapy approaches be possible?

- We appreciate the reviewer’s comment. We agree that in many countries including Germany, acupuncture is considered a standard treatment for chronic pain and musculoskeletal symptoms. The fact that acupuncture is covered by health insurance further proves the reviewer’s point. The specific “dual” system in Korea, on the other hand, divides the acupuncture from the other therapies such as surgery by healthcare system in which they are defined, categorized, and provided; both medical systems are reimbursed by National Health Insurance Service of Korea. The authors revised the manuscript and stated the difference as below:

Introduction, line 71-75:

Due to specific healthcare system in Korea, which provides dual healthcare divided into Korean Medicine and Western Medicine, acupuncture is provided by Korean Medical Doctors (KMD) who are licensed as Medical Doctors but specialize in diagnoses supplemented by medical knowledge based on Korean Medicine and therapies such as acupuncture, moxibustion, Tuina, and herbal medicine[16]. Although they are under the “dual” healthcare system, both are reimbursed by National Health Insurance Service of Korea. Korean medicine (KM) conservative treatments such as acupuncture and moxibustion have been reported to be safe and effective for short-term pain relief for lateral epicondylitis [6,17-24].

Method:

You describe possible risk factors for epicondylitis. Are post-traumatic epicondylitis also considered?

- We appreciate the reviewer’s comment. As the reviewer pointed out, we used M codes to define epicondylitis; while this includes chronic symptoms due to overuse with repetitive motions on the elbow such as the “Tennis elbow,” the definition did not include elbow injuries by trauma defined by S codes.

Based on the reviewer’s comment, the authors revised the manuscript as below:

This study was conducted on patients with lateral epicondylitis (KCD-10 code M771) as the primary diagnosis during the study period, and patients of all ages who had received healthcare service at least once with KCD-10 Code M771 as the primary diagnosis. While this included chronic symptoms on the elbow due to overuse with repetitive motions on the elbow such as the “Tennis elbow,” the definition did not include acute elbow injuries by trauma defined using S codes.

Results:

Section 3.1. is overloaded with numbers for the reader. Please revise.

We appreciate the reviewer’s comment and revised the manuscript as below:

According to Table 1, a total of 16,673 patients visited a healthcare facility for lateral epicondylitis in 2010, with 14,319 utilizing WM and 2,354 utilizing KM. In 2018, the total number of patients rose by about 45%, with a 39% increase of WM users and 87% increase of KM users. Further, the number of WM users was approximately 6 times greater than that of KM users in 2010, but the gap narrowed year after year to about 3.6-fold in 2014; however, the gap again widened since 2015 to approximately 4.5-fold in 2018. The total number of claims in 2010 was 63,649, with 53,704 claims in WM and 9,945 claims in KM. In 2018, the total number of claims was 96,301, with 76,256 claims in WM and 20,045 claims in KM. While the total number of claims and WM claims rose steadily, the number of KM claims rose until 2015 but began to decline since 2016. Total expenses rose by about 131% from 1,027,367 USD in 2010 to 2,377,540 USD in 2018. The WM expense and KM expense rose from 883,304 USD and 144,063 USD, respectively, in 2010 to 1,913,038 USD and 464,501 USD, respectively, in 2018. Per-patient expense in WM rose from 62 USD to 96 USD in 2018, and that in KM rose from 61 USD to 106 USD in 2018. Between WM and KM, the per-patient expense in WM was slightly higher than that in KM in 2010, but the per-patient expense in KM remained higher since 2011. However, the per-claim expense was higher in WM than KM every year during the nine-year period. The greater number of visits among KM users would have led to the greater annual per-patient expense in KM despite higher annual per-claim expense in WM.

Discussion:

Based on your results, can you make recommendations for guideline-based therapy? Please justify.

- We appreciate the reviewer’s question and revised the manuscript as below:

From a number of guidelines mentioned above, it seems reasonable to recommend Acu to treat lateral epicondylitis as guideline-based therapy; however, further studies are necessary to validate the cost-effectiveness of Acu in lateral epicondylitis.

Reviewer 2 Report

In their article, the authors have analyzed healthcare utilization for lateral epicondylitis in a nine year period. The manuscript is interesting. There are only some minor questions and issues.

Here are my comments:

  • Line 58: Remove „In“.
  • Line 64: Remove „and acupuncture“
  • The paragraph line 70-77 should be presented earlier since prevalence is described. Maybe before the paragraph dealing with diagnosis.
  • Introduction, line 84: What does HIRA stand for? Introduce abbreviation.
  • Line 103: What does PRP stand for? Introduce abbreviation.
  • Line 133: Is this the same as ICD code?
  • Results: The expenses in USD can be presented without cent values. This would make it easier for the reader.
  • Table 1: Please explain the abbreviation WM and KM in the table. Maybe following the reference to Supp Table 1)
  • Figure 3: Why is the number Figure 3 and not 1? Please correct.
  • Line 204: The number 92,258 does not correspond to the number in Table 2 (96,258).
  • Line 206: Add % after 52.22.
  • Lines 249-258: The comma is missing for the numbers (117,633.56)
  • Line 266: “…shows the five most…”
  • Line 298: Maybe use “second” instead of “next”

Author Response

Reviewer 2

Here are my comments:

Line 58: Remove „In“.

- We appreciate the reviewer’s comment and revised the manuscript as below:

Imaging is usually only performed when it is necessary to assess the severity of tissue injury and eliminate other causes.

Line 64: Remove „and acupuncture“

- We appreciate the reviewer’s comment and revised the manuscript as below:

Initial conservative treatments include behavioral modification, non-steroidal anti-inflammatory drugs (NSAIDs), strap, braces, physical therapy, extracorporeal shock wave therapy, injection, and laser therapy; and nonsurgical treatments are about 90% effective.

The paragraph line 70-77 should be presented earlier since prevalence is described. Maybe before the paragraph dealing with diagnosis.

- We appreciate the reviewer’s question and moved the paragraph to before the one with diagnosis:

The prevalence of lateral epicondylitis has been reported to range from 1–3% in the general population [16], and approximately one million cases are newly diagnosed every year in the United States [17]. In terms of age, people aged 40 years and over are more commonly affected [16]. In particular, the prevalence has been reported to be higher among people aged 40–49 years, followed by 50–59 years [17]. In a Finnish study on people aged 30–64 years, the prevalence was also higher among those aged 45–54 years [10]. In terms of sex, there were no marked differences between sexes [16], with the prevalence being reported to be 1–1.3% in men and 1.1–4.0% in women [13].

Introduction, line 84: What does HIRA stand for? Introduce abbreviation.

- We appreciate the reviewer’s comment and revised the manuscript as below:

Health Insurance Review and Assessment Service (HIRA)

Line 103: What does PRP stand for? Introduce abbreviation.

- We appreciate the reviewer’s comment and revised the manuscript as below:

Further, other studies investigated the effectiveness of extracorporeal shock wave therapy (ESWT), prolotherapy, platelet-rich plasma (PRP) injection, and arthroscopy

Line 133: Is this the same as ICD code?

- We appreciate the reviewer’s comment and revised the manuscript as below:

KCD code is the adopted version of ICD code in Korea with a few changes to reflect the clinical settings in Korea[18].

Results: The expenses in USD can be presented without cent values. This would make it easier for the reader.

- We appreciate the reviewer’s comment and revised the manuscript as below:

According to Table 1, a total of 16,673 patients visited a healthcare facility for lateral epicondylitis in 2010, with 14,319 utilizing WM and 2,354 utilizing KM. In 2018, the total number of patients rose by about 45%, with a 39% increase of WM users and 87% increase of KM users. Further, the number of WM users was approximately 6 times greater than that of KM users in 2010, but the gap narrowed year after year to about 3.6-fold in 2014; however, the gap again widened since 2015 to approximately 4.5-fold in 2018. The total number of claims in 2010 was 63,649, with 53,704 claims in WM and 9,945 claims in KM. In 2018, the total number of claims was 96,301, with 76,256 claims in WM and 20,045 claims in KM. While the total number of claims and WM claims rose steadily, the number of KM claims rose until 2015 but began to decline since 2016. Total expenses rose by about 131% from 1,027,367 USD in 2010 to 2,377,540 USD in 2018. The WM expense and KM expense rose from 883,304 USD and 144,063 USD, respectively, in 2010 to 1,913,038 USD and 464,501 USD, respectively, in 2018. Per-patient expense in WM rose from 62 USD to 96 USD in 2018, and that in KM rose from 61 USD to 106 USD in 2018. Between WM and KM, the per-patient expense in WM was slightly higher than that in KM in 2010, but the per-patient expense in KM remained higher since 2011. However, the per-claim expense was higher in WM than KM every year during the nine-year period. The greater number of visits among KM users would have led to the greater annual per-patient expense in KM despite higher annual per-claim expense in WM.

Table 1: Please explain the abbreviation WM and KM in the table. Maybe following the reference to Supp Table 1)

- We appreciate the reviewer’s comment and revised the manuscript as below:

WM, Western Medicine; KM, Korean Medicine.

Figure 3: Why is the number Figure 3 and not 1? Please correct.

- We appreciate the reviewer’s comment and revised the manuscript as below:

Figure 1.

Line 204: The number 92,258 does not correspond to the number in Table 2 (96,258).

- We appreciate the reviewer’s comment and revised the manuscript as below:

According to Table 2, more female patients visited a healthcare facility for lateral epicondylitis (96,258; 53.66%) than male patients (83,136, 46.34%).

Line 206: Add % after 52.22.

- We appreciate the reviewer’s comment and revised the manuscript as below:

the percentages were 47.78% and 52.22%, respectively.

Lines 249-258: The comma is missing for the numbers (117,633.56)

- We appreciate the reviewer’s comment and revised the manuscript as below:

The service category with the highest number of claims was physical therapy (117,633.56 cases)

Line 266: “…shows the five most…”

- We appreciate the reviewer’s comment and revised the manuscript as below:

Table 4 shows the five most frequently performed interventions in each service category in WM and KM.

Line 298: Maybe use “second” instead of “next”

 - We appreciate the reviewer’s comment and revised the manuscript as below:

The second most frequently performed intervention was infrared therapy

Round 2

Reviewer 1 Report

The authors addressed the comments. The manuscript has thus become even better. Thank you very much.